# Archaean and Proterozoic diamond growth from contrasting styles of large-scale magmatism

Janne M. Koornneef [1], Michael U. Gress[1], Ingrid L. Chinn[2], Hielke A. Jelsma [3], Jeff W. Harris[4] & Gareth R. Davies [1]

Precise dating of diamond growth is required to understand the interior workings of the early Earth and the deep carbon cycle. Here we report Sm-Nd isotope data from 26 individual garnet inclusions from 26 harzburgitic diamonds from Venetia, South Africa. Garnet inclusions and host diamonds comprise two compositional suites formed under markedly different conditions and define two isochrons, one Archaean (2.95 Ga) and one Proterozoic (1.15 Ga). The Archaean diamond suite formed from relatively cool fluid-dominated metasomatism during rifting of the southern shelf of the Zimbabwe Craton. The 1.8 billion years younger Proterozoic diamond suite formed by melt-dominated metasomatism related to the 1.1 Ga Umkondo Large Igneous Province. The results demonstrate that resolving the time of diamond growth events requires dating of individual inclusions, and that there was a major change in the magmatic processes responsible for harzburgitic diamond formation beneath Venetia from the Archaean to the Proterozoic.

[1] Vrije Universiteit Amsterdam, Faculty of Earth and Life Sciences, De Boelelaan 1085, 1081 HV Amsterdam, The Netherlands. [2] De Beers Exploration, Private Bag X01, Southdale 2135, South Africa. [3] Anglo American plc, Group Exploration and Geosciences, 45 Main Street, Johannesburg 2001, South Africa. [4] University of Glasgow, School of Geographical and Earth Sciences, Glasgow G12 8QQ, UK. Correspondence and requests for materials should be addressed to J.M.K. (email: j.m.koornneef@vu.nl)

The apparent antiquity of many diamond suites means that they represent a window to both the Earth's early geological record and deep carbon cycle[1]. Precise dating of diamond growth events is a prerequisite for such studies but represents a challenge owing to the complex growth history of many gem quality diamonds[2]. Inclusion-bearing diamonds derived from the sub-continental lithospheric mantle (SCLM), preserve evidence of tectono-thermal events such as continental assembly and mantle melting[1, 3–5]. Diamonds crystallise from metasomatic reactions involving C-H-O-S-rich supercritical-fluids and/or silicate melts and on rare occasions include minerals that carry the imprint of the environment of growth[2]. Diamonds, transported from the SCLM to the surface as xenocrysts in kimberlitic or related magma types, protect the mineral inclusions from secondary processes such as later mantle metasomatism and re-equilibration with the host magma[6]. Radiogenic isotope studies of inclusions in combination with compositional data can constrain the conditions and timing of diamond growth, and provide fundamental information about the tectono-magmatic processes that led to the formation and modification of the lithospheric keels that underlie the oldest parts of the Earth's continents[1]. Inclusions in diamonds derived from the sub-continental lithosphere are typically subdivided in three principal suites that characterise their mantle source rocks; peridotitic, eclogitic, and websteritic (see ref. [2] for an extensive review). The peridotitic suite is further subdivided into harzburgitic, lherzolitic, and wehrlitic parageneses based on Ca and Cr contents of garnet inclusions. Thus far, with one exception[7], the low trace element concentrations of peridotitic inclusions in diamond and their small size has necessitated pooling tens to hundreds of inclusions to obtain Sm-Nd isochron ages[8–13]. In consequence, fundamental concerns have been raised about common inclusion parentage and the validity of these ages as records of actual diamond growth events[14–16].

A previous study of inclusions from the Venetia diamond mine located within the Limpopo Mobile Belt in South Africa, examined 400 garnet inclusions of harzburgitic composition of which 140 were divided into four compositional groups for Nd

and Sr isotope analysis[13]. Three of the pooled groups defined a nominal Sm-Nd isochron age of $2.30 \pm 0.04$ Ga $(2\sigma)$ with an unradiogenic initial ratio ($\varepsilon$Nd = −8). The authors recognised that the results potentially recorded mixing of different components and presented a model wherein Venetia harzburgitic diamonds crystallised at ca. 2 Ga associated with modification of a > 3 Ga Archaean harzburgitic SCLM by Bushveld-related magmas. The implication of this interpretation is that harzburgitic garnets of highly variable composition were formed during a single diamond-forming event in the mid-Proterozoic.

To provide constraints on how the tectono-magmatic conditions responsible for harzburgitic diamond formation may have evolved over time, and assess the possible temporal evolution of the Earth's carbon cycle, we present coupled major and trace element and Sm-Nd isotope data for 26 individual garnet inclusions extracted from 26 individual peridotitic diamonds from Venetia, which were also measured for their carbon isotope compositions. The data allow the unravelling of the mixed signature recorded in previous pooled inclusion data, and produce new accurate ages that date regional, but contrasting, styles of magmatism that affected the mantle beneath the Limpopo Mobile Belt in the Archaean and Proterozoic. The data provide a tantalising glimpse of how processes associated with diamond formation may have changed over time.

## Results

**Genetic relations between inclusions and hosts.** The relationship between diamond and its inclusions is potentially key in interpreting absolute age information obtained from the inclusions. There has been significant recent debate as to whether inclusions are formed simultaneously with diamond (syngenesis), or are pre-existing objects incorporated during diamond growth (protogenesis)[17]. In addition, the term 'synchronous' has been proposed, describing protogenetic inclusions that owing to chemical equilibrium with the diamond-forming medium record the time of diamond growth[16].

All inclusions studied in this work show cubo-octahedral diamond imposed morphology and some show diamond inherited surface morphology (Fig. 1). The occurrence of imposed cubo-octahedral morphology on monoclinic (pyroxenes), hexagonal (monosulphides), orthorhombic (olivines), and cubic (chromites or garnets) inclusions ("negative crystals") has conventionally been taken as evidence that the two phases interacted during growth with the strong bonding of diamond imposing the cubo-octahedral morphology. Hence, inclusions with imposed morphology are considered to have formed syngenetically with diamond[18, 19]. Moreover, epitaxial relationships between inclusions and diamond hosts have also been thought as strong evidence of the control by diamond on the crystallisation of the inclusion[18–21]. Recent studies from Siberian diamonds reported that epitaxy played both a strong[22, 23] and limited control[24] on inclusion formation. Theoretical considerations of adhesion energy and interface energy, however, suggest that there is no energetic benefit for olivine-diamond interfaces to form an epitaxial relationship[25]. An X-ray topographic study of an olivine inclusion-bearing diamond reported an absence of volume distortion in diamond around the inclusions[17]. In addition, the surface morphology of olivine inclusions had stepped morphology. Both observations are consistent with growth or dissolution processes contributing to the final inclusion morphology. The recent report of epitaxy between clinopyroxene inside and outside a diamond in a lherzolite xenolith adds to the discussion[16]. These authors propose that the use of the term syngenetic may be inappropriate to describe inclusions. They argue that the term 'synchronous' is

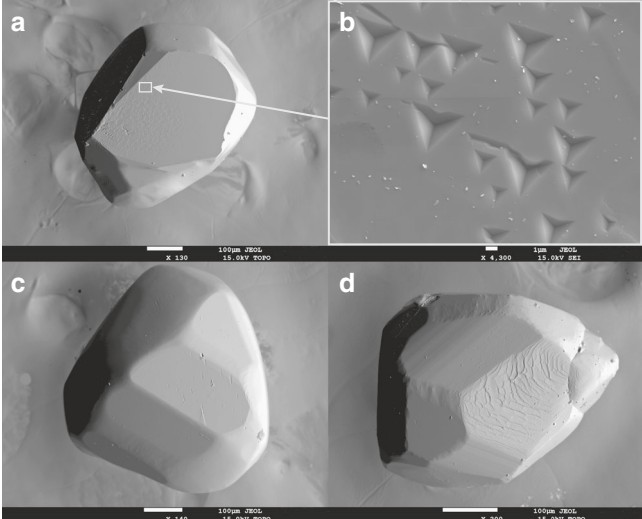

**Fig. 1** Electron microprobe images of garnet inclusions showing diamond imposed cubo-octahedral morphology and syngenetic growth features. **a** Garnet inclusion V471. The surface of V471 **b** has diamond-like "trigons" that establish syngenetic growth of inclusion and host. **c** Inclusion V445. The top surface of V405 **d** records stepped features and the side faces show well-developed growth lines consistent with syngenetic growth with the host diamond

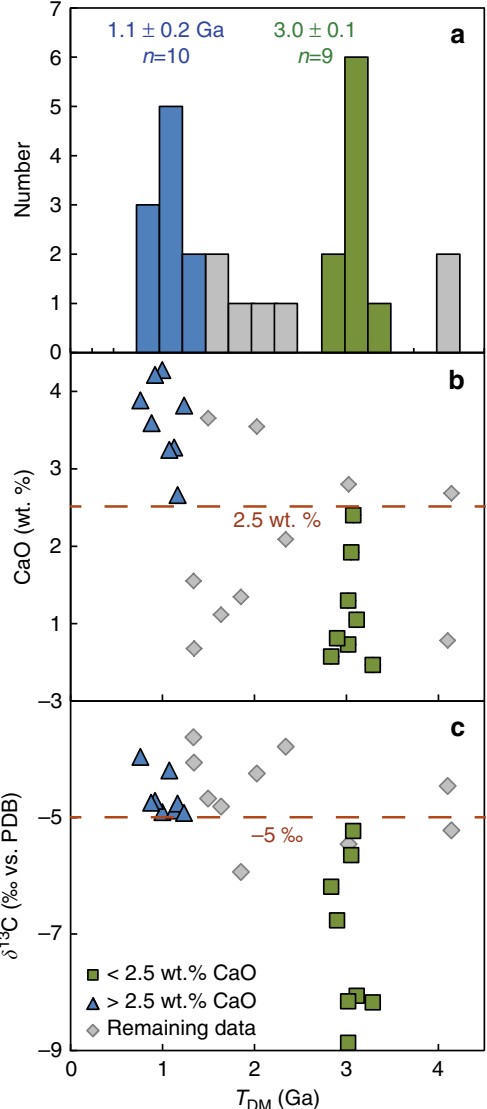

**Fig. 2** Nd depleted mantle model ages for 26 individual Venetia garnet inclusions and relations with CaO content and host diamond $\delta^{13}$C. **a** Depleted mantle model age ($T_{DM}$) histogram showing two distinct age groups; **b** The two age groups are further distinguished based on a CaO content of 2.5 wt.%; **c** The $\delta^{13}$C values of the host diamonds interiors for the older group of inclusions are lower (< -5 ‰) compared with the young age group (> -5 ‰)

more applicable to explain isotopic data when minerals were formed prior to diamond growth (protogenetic), but owing to chemical equilibrium with the diamond-forming fluid minerals will record the time of encapsulation in the diamond[16]. Experimental studies have produced syngenetic growth of inclusions from hydrous fluids containing salts, silicates, and carbonates under upper mantle conditions. Such experiments provide unequivocal evidence that diamonds and silicates can form syngenetically, producing mineral assemblages comparable to those found in inclusion suites in natural diamonds[26].

Taylor et al.[27], however, argued that the sinusoidal rare earth element (REE) patterns that characterise harzburgitic garnet inclusions in diamonds worldwide are indicative of a complex genetic history for the garnets prior to diamond encapsulation. The above authors argued that the sinusoidal REE pattern with low absolute heavy rare earth element (HREE) implies an extensive melt depletion phase in the absence of residual garnet

was followed by a metasomatic event that involved a highly fractionated light rare earth element (LREE) component, suggesting that the inclusion was protogenetic[27]. An equally plausible argument, however, is that the diamond-forming fluid interacted with the melt-depleted host peridotite to produce an environment with a sinusoidal REE pattern.

The interpretation of the above data remains a matter of debate. It appears definite that the external regions of inclusions with imposed morphology are syngenetic with diamond growth but that does not unambiguously prove that the bulk of the inclusions formed syngenetically[17]. Irrespective of the origin of the inclusions, a key aspect in dating a mineral inclusion is the extent to which the inclusion reached chemical equilibration with the diamond forming media, that is, do protogenetic grains record synchronous age information. The closure temperature of the Sm-Nd system in garnets in the crust and mantle varies depending on elemental diffusivity, mineral size, composition and cooling rate but is considered to be typically between 750 and 900 °C[28, 29]. In the case of a garnet in the mantle in contact with a metasomatic fluid/melt precipitating diamond, free diffusion of REE is thus expected at typical mantle temperatures (1000–1400 °C). Under such conditions, the time for a garnet of protogenetic origin to reach full Nd isotope equilibration with the fluid/melt will depend on the effective garnet grain size and elemental diffusivity. An upper limit of equilibration time can be determined by assuming a defect-free gem quality mineral under anhydrous conditions. Based on the typical size of our garnet inclusions (100 μm radius), full chemical equilibration of REE (Sm, Dy, Yb) with a diamond-forming melt/fluid would occur within less than 0.6 kyr at 1400 °C (diffusion data from ref.[29]). At a lower temperature of diamond formation (1000 °C) and again assuming a defect-free 100 μm radius garnet, it is possible that full chemical equilibration with the diamond forming melt/fluid could take up to 250 kyr. Hence under such circumstances, instantaneous diamond formation could potentially entrain minerals partially recording a protogenetic age. Given the diamond imposed morphology and growth features seen on the crystal faces of analysed inclusions (Fig. 1), we have, however, evidence that at least the outer portion of the garnet inclusions grew simultaneously with the diamond from the same fluid. Interaction and equilibration during dissolution/precipitation of garnet with the fluid is instantaneous, resulting in a syngenetic relationship[30]. Furthermore, experimental studies have shown that under hydrous mantle conditions, typical for diamond growth, garnet becomes more susceptible to dislocation creep, likely resulting in a marked increase in diffusion rates[31]. Hence, we conclude that retaining a protogenetic age, even in the mineral core, seems highly unlikely.

**Garnet inclusion geochemistry.** All inclusions analysed are harzburgitic garnets, 0.1–0.4 mm in size, with weights between 20 and 230 μg. Major element compositions of the inclusions were determined on unpolished, but flat and levelled crystal surfaces before trace element and isotopic analysis (see Methods section). CaO and $Cr_2O_3$ contents are variable, between 0.5 and 4.3 wt.%, and 5.1 and 17.0 wt.%, respectively (Supplementary Fig. 1 and Supplementary Data 1). The subcalcic compositions in equilibrium with a harzburgitic protolith are in agreement with previous work[32]. Garnet compositions record a broad negative correlation between CaO and Mg# that suggests that more melt-depleted garnets have lower CaO (Supplementary Fig. 1). The $^{143}Nd/^{144}Nd$ ratios (0.5097-0.5233) and Nd (0.9-26 ppm) and Sm (0.01–5.8 ppm) concentrations, used to calculate depleted mantle model ages ($T_{DM}$), are presented in Supplementary Table 2. Within a garnet harzburgite assemblage, the vast

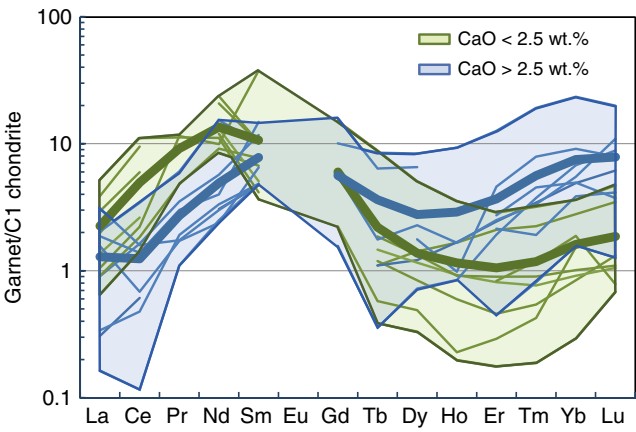

**Fig. 3** Chondrite normalised rare earth element patterns of samples in the two age groups. Green lines indicate 3.0 Ga samples and blue lines 1.1 Ga samples. Thick lines represent group averages. The 3 Ga group samples have sinusoidal patterns with higher LREE but lower HREE compared with the 1.1 Ga group samples

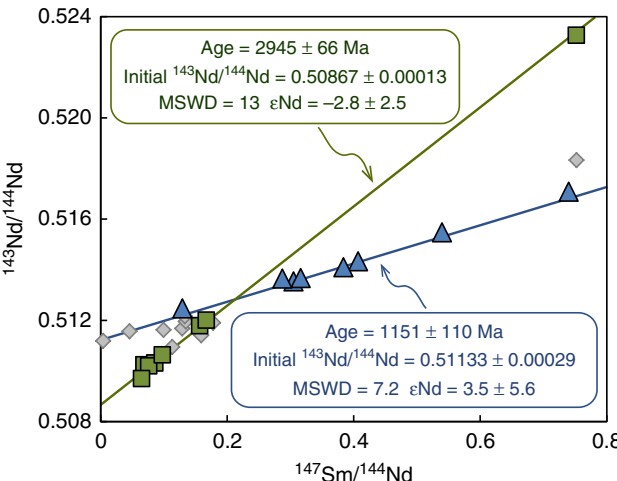

**Fig. 4** Sm-Nd isochron diagram for Venetia garnet inclusions. The two isochrons are distinguished based on $T_{DM}$ and CaO of the samples. Green square symbols represent the 3.0 Ga age group samples with lower than 2.5 wt.% CaO; blue triangles represent the 1.1 Ga age group with higher than 2.5 wt.% CaO. The analytical errors are smaller than symbol sizes.

majority of the REE will be contained in garnet ( > 90%), hence we can expect low Ca garnets to record geologically meaningful Nd model ages. This situation is in marked contrast to lherzolitic or eclogitic inclusion assemblages where element partitioning between clinopyroxene, garnet and the precipitating fluid/melt, will lead to fractionation of Sm/Nd ratios and most probably mineral model ages that can only be used for indicative purposes. Hence, the two populations of model ages derived from the harzburgitic garnets that can be distinguished in a probability density diagram are considered significant (Fig. 2a). An average $T_{DM}$ age of $3.0 \pm 0.1$ Ga is defined by nine samples and an age of $1.1 \pm 0.2$ Ga (1 SD) is recorded by 10 samples. The 3.0 Ga population generally has CaO contents lower than 2.5 wt.%, whereas the 1.1 Ga age population has CaO contents mostly higher than 2.5 wt.%. We use 2.5 wt.% CaO as a threshold value to further define the two groups, now with eight samples each (Fig. 2b). Significantly, the carbon isotope compositions of the host diamonds for the two age populations are also distinct; the older diamonds are depleted in $^{13}$C with $\delta^{13}$C $< -5$ ‰ ($-8.9$ to $-5.5$ ‰) and the 1.1 Ga age group has $\delta^{13}$C $> -5$ ‰ ($-4.9$ to $-3.6$ ‰, Fig. 2c, Supplementary Table 1). Chondrite-normalised REE patterns of garnets from the old and young groups also have distinct geometries. The older group of garnets has strong "sinusoidal" patterns with generally higher LREE abundances that increase from La$_N$ to Nd$_N$, has low Sm/Nd and concave down MREE-HREE. In contrast, the 1.1 Ga group has lower LREE abundances that increase from La$_N$ to Sm$_N$, has higher Sm/Nd and higher MREE-HREE abundances that define less-fractionated REE patterns. The minimum in the HREE is at Dy for the 1.1 Ga group and at Er for the 3.0 Ga group (Fig. 3).

Strikingly, the two inclusion populations define separate isochrons yielding ages of $2.95 \pm 0.07$ Ga and $1.15 \pm 0.11$ Ga. Both isochrons have initial ratios within error or close to chondritic mantle (Fig. 4, εNd of $-2.8$ and $+3.5$, respectively). No clear relationship that would indicate simple two-component mixing exists between $^{143}$Nd/$^{144}$Nd and reciprocal Nd (Supplementary Fig. 1d). The realistic initial ratios of the isochrons and the good agreement of the isochron ages with the depleted mantle model ages support the age significance of the isochrons. Seven additional inclusions give $T_{DMs}$ that are both older (4.1 Ga) and younger (1.3–2.3 Ga) than the 3.0 Ga age group (Fig. 2a). Even though some of these samples fall onto or close to the two well-defined isochrons (Fig. 4), their chemical characteristics

do not justify including them into the isochron populations. For example, two samples with $T_{DM}$ of 1.3 Ga (V306 and V471) plot on the isochron of 1.15 Ga at low $^{147}$Sm/$^{144}$Sm (Supplementary Table 2), however, their similar trace element patterns are distinct from the 1.15 Ga sample population (steep HREE, see Supplementary Data 1 and Supplementary Fig. 2), suggesting growth from a different growth medium. In addition, the two samples with $T_{DM}$ of 4.1 Ga have both low and high CaO (0.7; 2.7 wt%) but have relatively flat HREE patterns. They plot just below the 2.95 Ga isochron (Fig. 4), but again there is no justification for including them in the isochron.

## Discussion

Our data establish that it is vital to utilise single inclusions to obtain accurate age constraints on the time of diamond growth and that pooling of inclusions may mix populations of different age, resulting in ages with diminished geological meaning. The data reported by Richardson et al.[13] on combined harzburgitic garnet inclusions (32, 33, 34, and 42 specimens in each group) show a significantly smaller range in $^{147}$Sm/$^{144}$Nd and $^{143}$Nd/$^{144}$Nd compared with the individual inclusions, demonstrating that the pooling masked the original heterogeneity. More than 25% of the individual inclusions analysed here have higher Sm/Nd and $^{143}$Nd/$^{144}$Nd than the composite inclusion data, but because of low Nd contents their presence in a pooled population is obscured by inclusions with higher Nd contents. The average of the $T_{DM}$'s for individual inclusions (Supplementary Table 2) is identical to that obtained from the isochron of pooled garnet inclusions[13]. Richardson et al. concluded that harzburgitic diamond growth beneath Venetia was related to regional scale 'basaltic' magmatism of the Bushveld event at ~ 2 Ga. The younger diamond growth event at 1.1 Ga is characterised by garnets with relatively high Ca and Sm/Nd and hence is compatible with formation associated with large-scale regional basaltic magmatism, but significantly younger than inferred by the pooled inclusion data. The involvement of Archaean lithospheric mantle was inferred from the pooled inclusion data, however, the second harzburgitic diamond formation event at 2.95 Ga that produced a population of garnets with low Ca and low Sm/Nd remained unrecognised. These conclusions mean that

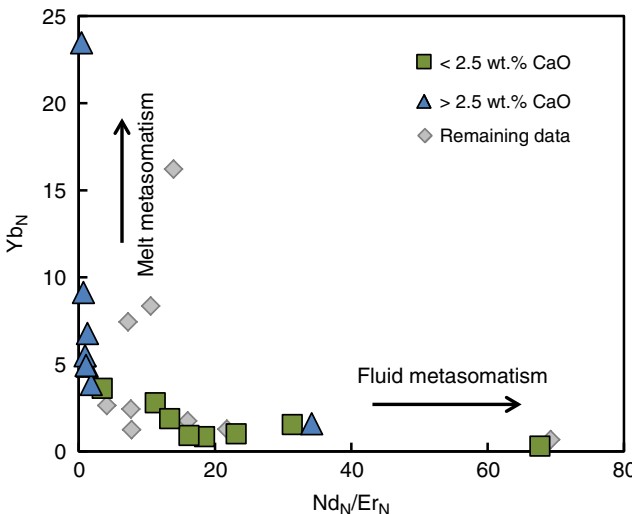

**Fig. 5** Chondrite normalised Nd/Er versus Yb for the harzburgitic garnets indicate two types of metasomatism. The low CaO samples with strongly sinusoidal REE$_N$ at low HREE document growth in a melt depleted reservoir that was metasomatised by a fluid. The > 2.5 wt.% CaO samples with modest HREE enrichment suggest melt dominated metasomatism[36].

Proterozoic ages ( ~ 2 Ga) obtained from pooled peridotitic garnet inclusions from Premier (Cullinan), SA[12] and Udachnaya, RU[33] may also be coincidental. Based on Re-Os dates from sulphide inclusions at Udachnaya, it has already been shown that old and young peridotitic diamond populations are recovered from these mines[34], further stressing the need to re-examine published diamond ages obtained from pooled diamonds.

Below, we consider the potential coupling between diamond growth within the lithosphere and large scale magmatic events expressed by the presence of crustal intrusions in the area with similar ages as the diamonds. A correlation is made between diamond formation and large tectono-magmatic events due to the large amount of carbon recovered in the form of diamond from the Venetia mine. The Venetia kimberlite cluster comprises a surface area of only 28 ha but since its opening in 1992 has produced in the order of 100 M carats and comparable resources remain to be exploited (Anglo American Annual Reports and[35]). This equates to > 40 tons of C moved from the mantle to the upper few km of the crust with a far larger mass of C stored in the kimberlites at depth and lost due to erosion of the upper levels of the kimberlite.

Kimberlites represent small degree melts that entrained material from only limited lateral extent within the SCLM. The majority of inclusions at Venetia are harzburgitic and derived from over a depth range of ~ 100 km[12]. Hence we conclude that diamond formation must be related to large scale tectono-magmatic events.

The different major and trace element geochemistry of the garnet inclusions studied here establishes diamond growth under distinct conditions separated by almost two billion years. Fluid and melt dominated metasomatism result in distinctive trace element signatures with highly to mildly incompatible element contents (for example, LREE/HREE) decreasing strongly from fluids to melts[36–38]. In a diagram that quantifies the sinuosity of the REE pattern (for example, Nd$_N$/Er$_N$) the two garnet inclusion groups show distinct behaviour related to fluid or melt dominated metasomatic diamond growth; (Fig. 5). The older harzburgitic garnets with low CaO, low HREE and sinusoidal REE patterns can be related to low-$T$ fluid dominated metasomatism. Given the

inclusion compositions, the metasomatic fluid was likely low in Ca, Fe, Ti, and Al, a characteristic of melt-depleted residua. The LREE enrichment in the garnet inclusions, however, indicates involvement of a C-H-O medium enriched in trace elements that reintroduced LREE to the depleted SCLM. In contrast, the younger garnet inclusions have elevated CaO, FeO and HREE. This type of metasomatism is typically associated with high-$T$ silicate melt-metasomatism that forms lherzolitic type garnet inclusions when pervasive[36] and is interpreted to be related to melts derived from the asthenosphere.

Based on the regional geological evolution of the Zimbabwe Craton, we infer that the older, 2.95 Ga diamonds formed in a highly melt-depleted mantle residue from fluids mobilised by passive asthenospheric decompression during crust-forming magmatism (Fig. 6). The diamond growth event predates the Limpopo Mobile Belt that formed from collision of the Archaean Kaapvaal and Zimbabwe Cratons at ~ 2.65 Ga[39]. Key to linking the diamond-formation event to magmatism in the region is reconstruction of the crust-mantle architecture prior to assembly of the two cratons. The origin of the Central Zone of the Limpopo Belt, the location of the Venetia diamond mine, is debated[40]. De Wit et al.[41], for example, argue using geophysical data that the SCLM beneath the Central Zone of the Limpopo Belt was part of the Zimbabwe Craton with the overlying crust belonging to the Kaapvaal Craton or to an allochtonous block that over-thrusted towards the north[39, 40, 42, 43]. The low temperature peridotites from Venetia, however, show marked Si-enrichment more characteristic of the Kaapvaal SCLM, in contrast to the melt depleted diamond facies samples[44]. These data suggest potential decoupling of shallow and deep peridotite suites, a conclusion compatible with mantle-crust decoupling, which is supported by lead isotope data and the inferred derivation of crustal rocks from a common source with a long-lived high μ (U/Pb) value in the Central Zone and on the Zimbabwe Craton. This Pb isotope signature differs from the inferred low μ source below the Southern Limpopo Belt and the Kaapvaal Craton[45]. Based on the suggested decoupling, we expect the crustal rocks associated with the 2.95 Ga diamond growth event at Venetia to be located to the north in the Zimbabwe Craton. The southern part of this Craton contains multiple crustal units dated at 2.9–3.0 Ga[46–48]. The Chingezi Tonalites in the Belingwe greenstone belt, for example, yielded whole rock Sm-Nd ages between 2.95 and 3.05 Ga with an average initial $^{143}$Nd/$^{144}$Nd of 0.50884[47], within error of our 2.95 isochron initial ratio (0.50867 ± 13). Similarly, zircons from the Mashaba Tonalite, part of the Early Archaean Tokwe region (Fig. 6), yielded crystallisation ages for major crustal accretion between 2.8 and 3.05 Ga[49]. Both groups of intrusions are inferred to represent the plutonic equivalents of contemporaneous felsic volcanic rocks in the Belingwe and Buhwa greenstone belts[50, 51]. The tectonic regime at this time is interpreted to have changed from a continental magmatic arc (Tokwe Terrain) to a rifting passive continental shelf associated with asthenospheric decompression and melting of the mantle beneath the lithosphere[51]. The resulting magmatism across the region is inferred to have driven the fluid-dominated metasomatism at depth within the depleted lithosphere that was responsible for diamond growth at 2.95 Ga.

Importantly, the host diamonds of the Archaean inclusions generally have δ$^{13}$C lower than −5‰ with a significant spread (average = −7.1 ± 2.7‰ vs. PDB, 2 SD). The isotopic variation defines a broad positive relation with CaO in the garnet inclusions (Supplementary Fig. 1c). The most subcalcic garnets have diamond hosts with the lightest carbon and vice versa. Notably, the garnets do not have typical subduction signatures, with Nb/La > 1 (Supplementary Data 1 and ref. [38]). Thus, these δ$^{13}$C values are unlikely to involve a recycled organic carbon

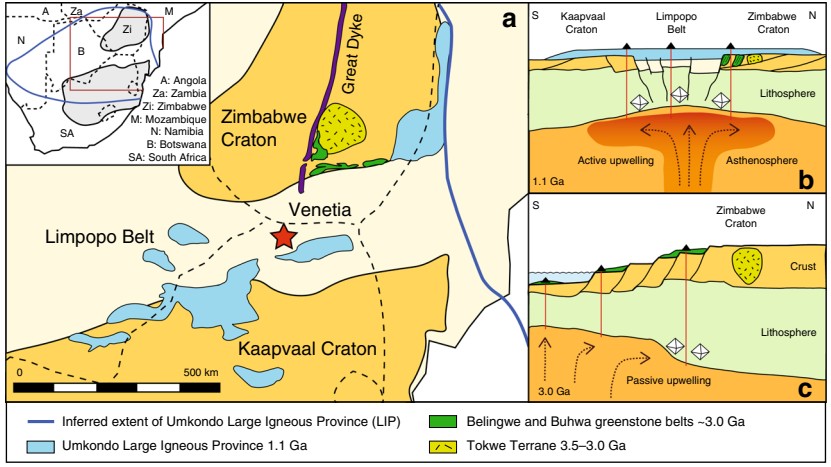

**Fig. 6** Simplified geological map and cross-section cartoons. **a** Shows magmatic rocks in the region that result from large scale tectono-magmatic events, inferred to be related to diamond growth beneath Venetia (red star). Dashed black lines in **a** are international boundaries. Outlines of Umkondo LIP outcrops are after ref. [57]. **b** Suggested tectonic setting at 1.1 Ga during formation of the Umkondo LIP. **c** Inferred setting at the 3.0 Ga diamond growth event. **b**, **c** are not to scale

component[52, 53]. The variable $^{13}C$ depletion of this fluid event is considered to reflect Raleigh-style fractionation suggesting diamond formation from relatively small volumes of super-critical fluids. The range in $\delta^{13}C$ values implies diamond precipitation involving oxidation of a reduced fluid ($CH_4$) causing isotopic fractionation to lower $\delta^{13}C$[54, 55]. The observed relation between CaO and $\delta^{13}C$ possibly reflects the effect of decreasing solubility in reduced C-saturated fluids moving upwards along a geothermal gradient[56].

The $1.15 \pm 0.11$ Ga diamond growth event can be linked to the Umkondo Large Igneous Province (LIP) magmatism that occurred between 1.106 and 1.112 Ga over an area of 2 million km² across the Kaapvaal and Zimbabwe Cratons[57] (Fig. 6). Compositions of remnant sills and dykes are homogenous over the region suggesting formation from a large degree melting event in the asthenosphere[58]. The major and trace element compositions of the younger garnet inclusions imply an origin related to high-$T$ melt metasomatism[38] (that is, high HREE are only recorded in samples with inferred temperatures > 1190 °C). This observation is consistent with melt infiltration into the depleted lithosphere caused by impingement of the thermal anomaly on the base of the SCLM that led to the Umkondo LIP. The $\delta^{13}C$ values of the 1.15 Ga diamonds record a restricted range with an average of −4.6 ‰, indistinguishable from asthenospheric carbon[1]. In addition, the initial ratio of the 1.15 Ga isochron (εNd + 3.5 ± 5.6) is in agreement with derivation from the convective depleted mantle.

The Venetia data indicate that Proterozoic diamond formation was associated with melt metasomatism resulting from a major thermal perturbation of the SCLM by active asthenospheric upwelling related to a LIP event, whereas Archaean harzburgitic diamond formation can be related to relatively cool fluid-dominated metasomatism caused by rift-related magmatism. Resolving whether the observed change in the tectonic processes responsible for harzburgitic diamond growth is a consequence of planetary-wide temporal evolution requires detailed dating of more diamond inclusion suites. Data presented here demonstrate that dating by pooling of garnet inclusions in diamond may result in averaging of ages from multiple diamond generations and potentially obscure the actual geological events responsible for diamond formation. This conclusion means that ages obtained from pooled peridotitic garnet inclusions from Premier (Cullinan), SA[12] and

Udachnaya, RU[33] may also be incorrect, bringing into question an apparent increase in worldwide peridotitic diamond formation at ~ 2.0 Ga[2]. Our data provides convincing evidence for at least two distinct diamond growth episodes beneath Venetia. Seven samples with $T_{DM}$ both younger and older than the 3.0 Ga age group may reflect additional diamond growth episodes. Further diamond and mineral inclusion investigations are required to assess whether the observed change in the tectonic processes responsible for the distinct harzburgitic diamond growth events at Venetia, is a continent- or even a planetary-wide occurrence.

## Methods

**Sample selection and preparation.** The studied garnet inclusions and host diamonds were donated to J.W.H. by the Diamond Trading Company, a member of the DeBeers Group. The inclusions had been liberated from their host diamonds by J.W.H and co-workers in the 90s[59]. The harzburgitic garnet inclusions studied here were selected primarily on the basis of their relative large size to allow precise and accurate Sm-Nd isotope analyses.

Major element compositions were determined on flat unpolished but levelled garnet crystal faces by electron microprobe (EMPA, JEOL JXA 8530F Field Emission Probe Microanalyser, at Utrecht University) following the methods of Timmerman et al.[60]. After EMPA analyses, carbon coating was removed by washing in ultrapure methanol and milli-Q water. The inclusions were weighed on a Mettler 7 decimal balance with a sensitivity of 0.1 μg and spiked with a $^{149}Sm$-$^{150}Nd$ mix aiming for a Nd sample/spike ratio of 20 with Nd concentrations estimated based on the $Cr_2O_3$-CaO compositions[13]. Samples were dissolved in a concentrated $HF$-$HNO_3$ mix on a hotplate at 140 °C for a minimum of 3 days. Chemical separation of Nd and Sm from the dissolved matrix was performed in a two column procedure using 0.16 ml TRU-resin and 0.7 ml LN resin[61]. Matrix fractions were collected and combined for trace element analyses by inductively coupled plasma mass spectrometry (Thermo X-Series II) at the Vrije Universiteit (VU) Amsterdam. BHVO-2 ($n = 5$) was used to correct for the chemistry yields for each element. Three inclusions, too small to date, were dissolved and analysed for trace elements directly by ICPMS. Data from these extra inclusions overlap both the SIMS data for garnet inclusions from Stachel et al.[38] and the dated inclusions processed through chemistry, thus confirming the yield corrections to be appropriate (Supplementary Fig. 2).

**Isotope analyses by thermal ionisation mass spectrometry.** Sm and Nd were analysed by Triton Plus using four $10^{13}$ Ohm amplifiers installed in the instrument since 2012[62]. Faraday-amplifier gains were determined using the La Jolla Nd standard following the method of Timmerman et al.[63]. Beams on $10^{13}$ Ohm amplifiers where kept below $3 \times 10^{-13}$ A. Samples were run to exhaustion. Instrumental reproducibility for 100 pg samples is determined by routine analysis of an in-house Nd standard (CIGO) that yields $0.511332 \pm 67$ ($n = 49$). Long-term reproducibility of 100 ng loads of this standard measured on $10^{11}$ Ohm amplifiers, yields $0.511334 \pm 10$ ($n = 28$), which equates to 0.511834 for La Jolla. Isotope composition and isotope dilution Nd and Sm concentrations were determined using an iterative approach to strip the spike and perform an exponential

instrumental mass fractionation correction ($^{146}Nd/^{144}Nd$ of 0.721903 for Nd and $^{147}Sm/^{152}Sm$ of 0.56081 for Sm). Small USGS BHVO aliquots ( ~ 1 mg) processed along with the inclusions yielded average Sm and Nd concentrations of 6.026 ± 0.038 and 24.34 ± 0.66 ppm, respectively, and $^{143}Nd/^{144}Nd$ = 0.512956 ± 0.000074 (2 SD, n = 7). Total procedural Sm and Nd blanks averaged 0.4 and 0.3 pg (n = 5). Blank corrections using the isotopic composition of lab solutes (0.511856 ± 90) have a negligible effect on isotope ratios and calculated $T_{DM}$, and were not performed. Depleted mantle model ages were calculated with the parameters of Michard et al.[64].

**Carbon isotope analyses of host diamonds.** Carbon isotope ratios were determined on diamond fragments using a Carlo Erba NC 2500 element analyser coupled to a Thermo Finnigan Delta Plus isotope ratio mass spectrometer at VU, Amsterdam. Potential drift was monitored every six samples using pairs of internal standards (WK synthetic diamond powder, −7.16‰ and natural diamond powder supplied by P. Cartigny, −8.22‰). External standards USGS24 (−16.05‰) and VICS ( + 1.35‰) were measured in triplicate with each batch of 30 samples. Fragments of host diamonds were selected based on morphology, (external surfaces and interiors) and in four cases the specific growth zone that housed the inclusion was identified. Interiors and external surfaces of diamonds have consistent $\delta^{13}C$ except for one diamond (see Supplementary Table 1). Further details of the stable and radiogenic isotope analytical protocols can be found in Timmerman et al., EPSL 2017[63].

**Data availability.** All data presented in this manuscript are available as Supplementary Information (Supplementary Data 1 and Supplementary Tables 1, 2).

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

## Acknowledgements

The research leading to these results has received funding from the European Research Council under the European Union's Seventh Framework Programme (FP7/2007-2013)/ ERC grant agreement no. 319209. J.M.K. received funding from the European Union's Horizon 2020 research and innovation programme under grant agreement No 654208 (Europlanet 2020 RI). J.W.H. thanks the Diamond Trading Company (a member of the De Beers Group of Companies) for the donation of the diamonds used in this study. We thank Richard Smeets, Bas van de Wagt and Sergei Matveev for support in the VU and UU laboratories and Suzette Timmerman for data discussion.

## Author contributions

J.M.K. performed the majority of the lab work and wrote the first version of the paper. All authors were involved in extensive discussions of data interpretation and edited the manuscript. M.U.G. and G.R.D. performed part of the major element and carbon isotope analyses. J.W.H. selected and liberated the inclusions and provided the samples.

## Additional information

**Competing interests:** The authors declare no competing financial interests.

**Reprints and permission** information is available online at http://npg.nature.com/ reprintsandpermissions/

