## [Peer Review File · Nature Communications]

Reviewers' comments:

Reviewer #1 (Remarks to the Author):

Overview

This is a very impressive paper that uses state-of-the-art isotopic measurements to date individual garnet inclusions in diamonds in a way that has not previously been possible. The results are strikingly different from a previous study of samples from the same locality where inclusions were pooled in order to obtain enough material for analysis. The data are used to draw conclusions about history of diamond formation in this part of southern Africa and to relate the diamond growth episodes to other regionally important magmatic events.

Detailed comments

1. I think the name of the locality should be included in the title
2. I agree with the views expressed in the discussion of syngenetic vs synchronous growth and although I have not done the calculations I agree that probably, even if protogenetic, the inclusions would re-equilibrate with a diamond-forming fluid to give the correct trapping date. However I am puzzled by the repeated claim to support this idea that elemental diffusion in volatile-rich melts-fluids is essentially instantaneous. Surely more important in the case of protogenetic crystals is the diffusion rate of Sm and Nd in garnet. If the authors wish to claim that protogenetic or syngenetic inclusions would give the same date they need to attempt some modelling of REE diffusion in garnet.
3. I found the symbols in figure 2 a little confusing. The top panel has 10 blue coded inclusions and 9 green coded inclusions, but panel 2 and 3 have only 8 blue and 8 green. There are 26 inclusions in each panel so some inclusions of each of the age peaks are converted to grey in panels 2 and 3. The justification for this is not clear to me.
4. I would like to see some discussion of the samples that do not fall within the 2 main age peaks. Are these bad data? Or was diamond formation going on all the way from 4 Ga to 1 Ga?
5. Although the single inclusion data is very impressive it is a shame that there is not more information on the growth zones from which the inclusions were extracted. The latter approach was done, very effectively, by Suzette Timmerman from the same group. Simply using the surface features of the diamond to classify as inside or rim (Table S2) is rather crude. By losing the context of the inclusions it is not clear whether each individual diamond is from a single growth event (as implicitly assumed here) or whether there are diamonds with cores and rims of different ages. Diamond V683 has very different $\delta^{13}\text{C}$ for inside and rim implying that this sample may have a core and overgrowth of very different ages. FTIR on the fragments would have been helpful (see point 6).
6. I find it fascinating that the 2 periods of growth identified here by radiometric dating are remarkably similar to the dates of growth deduced by FTIR for Murowa, if it is assumed that the temperature experienced by Murowa diamonds was constant throughout their history in the mantle (note the important caveat!). In our recent paper on FTIR techniques we used a Murowa diamond as an example of 2 stage thermal modelling (Kohn et al., *Lithos* 265 148-158). The scenario explored in our figure 8 has initial diamond growth at 3.2 Ga, an overgrowth at 1.1 Ga and exhumation at 500 Ma. This is remarkably similar to the scenario proposed here for Venetia and considering that the two localities are only 200-300 km apart I wonder if they are both affected by the same regional events.

Simon Kohn
University of Bristol

Reviewer #2 (Remarks to the Author):

Review of NCOMMS-17-03711-T
Nature Communications

Title: Archaean and Proterozoic diamond growth from contrasting styles of large-scale magmatism

Authors: Janne M. Koornneef, Michael U. Gress, Ingrid L. Chinn, Hielke A. Jelsma, Jeff W. Harris, and Gareth R. Davies

The manuscript reports dates derived from individual garnet inclusions from 26 diamonds, sourced from Venetia, South Africa. Sm-Nd model ages derived from individual garnets represents a relatively new technological capability for isotopic dating. In recognizing two model age clusters, the authors divide the garnets into an older 3 Ga group and a younger 1.1 Ga group, which are further distinguished by their Ca content and the carbon isotopic composition of the host diamond. The two groups also have slightly different rare-earth element patterns. One of the more compelling observations is that the two groups of garnets each appear to conform to well-behaved Sm-Nd isochrons.

The authors stress that a previously reported isochron age obtained from measuring composite/pooled garnet suites (Richardson et al. 2009) is incorrect. From here the connection is drawn between the two newly-defined age groups (2.95 Ga and 1.15 Ga) and contemporaneous large-scale crustal magmatic events. Trace element characteristics of the two groups are used to argue that the 2.95 Ga garnets record diamond growth from "cool" C-O-H fluids, while the 1.15 Ga garnets record diamond growth related to melt metasomatism.

Overall, the resulting data are of good quality and the implications are interesting. However, the presentation of the derived ages (especially the isochrons) could be strengthened and the large-scale implications require further development. The case to back up the isochrons could be made more thoroughly, which would make for a stronger paper. It would also be helpful to make some brief mention or interpretation of the 7 garnets whose model ages don't fit nicely into the two modes.

Currently too much discussion is devoted to pointing out the "incorrect" age obtained from pooled garnet analyses (Richardson et al. 2009). These previous measurements were made quite meticulously and are not really "incorrect," so much as they are misinterpreted. It is obvious that pooling samples can give an average measurement, but these previous measurements are still in total agreement with the new data and can now be re-interpreted more accurately. More could be gained by embracing the old data rather than refuting it.

The main area with room for improvement lies in the connection between diamond formation and tectonic/magmatic activity in the crustal record. The title and abstract make such claims boldly, but the text falls short of making a concrete connection. The parallels between garnet inclusion ages and large-scale magmatism are oversimplified. Nevertheless, the data and the potential for understanding large-scale processes recorded at the top (crust) and bottom (diamonds) of the lithosphere are very intriguing. If a case is to be made for strong connection between diamond formation and large-scale magmatism, a more detailed discussion, with geological map and perhaps cartoon model, should rightly be included.

More specific comments:

-Since line numbers were not included, the comments below will refer to page number, and line number counting down that particular page.

Title (p1): A title revision is suggested, following revisions to the manuscript. It may be more helpful to note the actual numerical ages determined, and specifying garnet inclusions and the locality.

P2, line 4: state the number of diamond samples

P2, line 9: consider changing two billion to 1.8 billion to avoid giving the reader the appearance of exaggeration

P2, line 12: consider changing "incorrect" to a different word, such as a "misinterpretation", since the previous data are still correct. The incorrect part was the underpinning assumption that the collection of Venetia peridotitic diamonds were of a single age.

P2, line 13: missing "The" before "results"

P3, line 4: Here it is stated that diamonds form from supercritical C-H-O-S fluid, which is consistent with popular ideas in the current literature, but later in the text diamond growth from melt is discussed, which seems inconsistent. Granted, this is not fully resolved in the literature, but as it is written, this inconsistency might be confusing to readers.

P3, line 4: remove comma

P3, line 6: it is important not to label kimberlites as the sole bearer of diamonds, since related volcanic rocks such as lamproite also transport diamonds

P3, line 5-7: Here and in a few other places, sentences need some grammatical nit-picking (Here the verbs are not in parallel form: "are transported" versus "protect")

P3, line 9: "fundamental information about the evolution of the Earth's interior" sounds a little vague

P3, line 10: add "lithospheric" before "diamonds"

P3, line 14: be more specific than "low elemental abundances" (trace element concentrations?)

P3, line 19-20: "a previous study... studied"

P3, line 21: change "combined" to "divided"

P4, line 3: change "conclusion" to "interpretation"

P4, line 10: the assertion that the inclusions "directly date" magmatism might be a bit too bold. Consider just "date"

P4, line 13: consider change "processes controlling diamond formation" to "processes associated with diamond formation"

P4, line 16: consider adding an average inclusion diameter, since the weight is not easy for most readers to interpret or visualize (e.g. 0.2-0.4 mm garnets)

P4, line 21: This statement only mentions fluid. What about the diffusion through the solid minerals? Reader may wonder if diffusion from fluid-mineral interface into the interior of a solid

mineral is also rapid. Is the garnet above “closer temperature”?

P5, line 1: specify if the flat crystal surface was level (i.e. ideally surface is both flat and perpendicular to beam)

P5, line 1: delete comma

P5, line 6: new sentence doesn't flow from previous sentence

P5, line 13-14: carbon isotope values are missing minus signs

P5, line 16: consider changing “sigmoidal” to “sinusoidal”

P5, line 17-18: description of REE pattern shape could be more clearly written

P5, line 19: the isochrons are impressive and could be discussed in more detail and mentioned in abstract

P5, line 20: Consider changing “no relationship” to “no clear relationship that would indicate simple two-component mixing”. There is still some discussion here that should be done to increase the reader's confidence in the isochrons.

P6, line 2-16: This section can be shortened significantly. Also, the mention of Premier and Udachnaya pooled ages should appear here in this section rather than in the final sentence of the paper.

P6, line 4: consider changing “incorrect” to “average” or “ages with little real meaning”

P6, line 8: change “published data” to “composite inclusion data” or “pooled inclusion data”

P6, line 12: the association between 1.1 Ga diamond growth and “basaltic” magmatism appears abruptly here, for the first time, without rationale

P6, line 15: consider changing “unconstrained” to “unrecognized”

P6, line 19: “The different geochemistry of garnet inclusions” should be worded more clearly to specify the two age groups

P6, line 22: it would be helpful to explain briefly why Fig 5 indicates melt or fluid dominated metasomatism. This is important in developing the rest of the story and making connections to magmatism.

P6, line 23: “low HREE” –should this say “high HREE”?

P7, line 3: why does LREE enrichment indicate so specifically a “C-H-O medium highly enriched in trace elements”?

P7, line 5: “elevated ... HREE” –should this say “lower HREE”?

P7, line 7: “melts derived from the asthenosphere” sounds vague

P7, line 10: “fluids mobilized by crust-forming magmatism” sounds vague

P8, line 2: More explanation is needed here in the inferred connection between contemporaneous tonalite plutons and diamond growth.

P8, line 3-6: consider rewording this sentence to make it easier to read

P8, line 7: sentence truncated at "and"

P8, line 9-10: Is this fractionation mechanism valid in light of recent work suggesting that mantle harzburgite is unable to buffer significant O₂ required support redox-reaction diamond formation models in the SCLM? (Luth, R.W., Stachel, T., 2014. The buffering capacity of lithospheric mantle: implications for diamond formation. *Contrib. Mineral. Petrol.* 168, 1083.)

P8, line 12: How is diamond formation directly linked to the Umkondo LIP? Is their age overlap and broad mantle-derived geochemical similarity enough to say they are directly linked? More justification and explanation would be helpful here.

P8, line 17: Explain briefly why the garnets "imply an origin related to high-T melt metasomatism"

P8, line 23: Explain why a "major thermal perturbation" is required

P9, line 2: Why is the magmatism claimed to be "rift-related" here? Rifting was not mentioned earlier in the discussion of tonalite plutons

P9, line 2-4: This sentence could use clarification

P9, line 6: change "obscures" "obscure"

Figure 1: This figure is not absolutely necessary and could be moved to supplement, making room for some kind of geological map/model.

Figure 2: Should the legend/symbol for "all data" be called "remaining" or "unassigned" or "outlier" data? The collection of grey diamonds is not all the data. Same goes for Fig 5.

Figure 2 caption: Consider changing "more depleted" to "lower." Are the carbon isotope values shown average measurements for each diamond?

Figure 3 caption: The caption says blue is the 3.0 Ga group, but the legend says blue is the high Ca group (and green is 1.1 Ga, low Ca). This does not agree with Figure 2.

Figure 4 caption: Refers to symbols in Fig 1, but shouldn't it say Fig 2?

References, ref 19: incomplete

Supplementary Information:

-How were inclusions chosen for the study?

-How were the inclusions removed from the diamonds?

-Some physical description of the diamonds themselves should be given. If there are really two major age groups, it would be interesting to note if there are any differences in the diamonds, aside from the carbon isotopes (e.g. morphology, nitrogen concentration, nitrogen aggregation, internal growth zonation).

P2, line 9: do not abbreviate concentrated to "conc."

P4, line 5: change "inclusions formed" to "inclusions are formed"

P5, line 7: should this refer to Fig S3 rather than Fig 2?

P6, line 2: "Here we date individual garnet inclusions..." this sounds out of place, like it belongs in the introduction to the main text

Figure S2 caption: For symbols shouldn't it say refer to Figure 2? For the "small grey diamonds," consider changing to a different symbol, since grey diamonds are already used, and mention here that these additional data points are for Venetia garnets.

Figure S3: this figure is not referred to in the text

Table S1: state units (wt %, ppm)

Table S3: provide reference for Venetia age of 520 Ma

Reviewer #1

Overview

This is a very impressive paper that uses state-of-the-art isotopic measurements to date individual garnet inclusions in diamonds in a way that has not previously been possible. The results are strikingly different from a previous study of samples from the same locality where inclusions were pooled in order to obtain enough material for analysis. The data are used to draw conclusions about history of diamond formation in this part of southern Africa and to relate the diamond growth episodes to other regionally important magmatic events.

We are pleased with Simon Kohn's overall positive and constructive comments and have followed almost all suggestions in the text and or figures. We acknowledge Simon for his input at the end of the manuscript.

Detailed comments

1. *I think the name of the locality should be included in the title*

We have added "beneath Venetia, South Africa" to the title, which now reads:

"Archaean and Proterozoic diamond growth from contrasting styles of large-scale magmatism beneath Venetia, South Africa"

2. *I agree with the views expressed in the discussion of syngenetic vs synchronous growth and although I have not done the calculations I agree that probably, even if protogenetic, the inclusions would re-equilibrate with a diamond-forming fluid to give the correct trapping date. However I am puzzled by the repeated claim to support this idea that elemental diffusion in volatile-rich melts-fluids is essentially instantaneous. Surely more important in the case of protogenetic crystals is the diffusion rate of Sm and Nd in garnet. If the authors wish to claim that protogenetic or syngenetic inclusions would give the same date they need to attempt some modelling of REE diffusion in garnet.*

The reviewer is correct that the REE equilibration of a potential protogenetic garnet is controlled by the diffusion rate of elements within the garnet. The closure temperature of the Sm-Nd system in gem quality garnets varies depending on mineral size, composition and cooling rate but is considered to be typically between 750 and 900°C (Van Orman et al., 2002; Ganguly and Tirone, 2009). In the case of a garnet in the mantle in contact with a metasomatic fluid/melt precipitating diamond, free diffusion of REE is expected at typical mantle temperatures (1000-1400 °C, , e.g., Van Orman et al., 2002). Under such conditions, the time for a garnet of protogenetic origin to reach full Nd isotope equilibration with the fluid/melt will depend on the effective garnet grain size. An upper limit of equilibration time can be determined by assuming a defect-free gem quality mineral under anhydrous conditions (both unlikely assumptions). Based on the typical size of our garnet inclusions (100 um radius), full chemical equilibration of REE (Sm, Dy, Yb) with a diamond forming melt/fluid would occur within less than 0.6 kyr at 1400 °C (diffusion data from Van Orman et al., 2002). At the temperatures estimated for diamond formation (1000 °C) and again assuming 100 um radius, it is possible that full chemical equilibration with the diamond forming melt/fluid could take up to 250 kyr. Hence under anhydrous conditions and assuming gem quality minerals, instantaneous diamond formation could potentially entrain minerals partially recording a protogenetic age. However, given the imposed cubo-octahedral morphology and the surface growth features seen on the inclusions crystal faces we have evidence that at least the outer portion of the garnet inclusions grew simultaneously with the diamond from the same fluid. Interaction and equilibration during dissolution/precipitation of garnet with the fluid is instantaneous resulting in the observed syngenetic relationship. Hence the above calculations demonstrate that retaining a protogenetic age, even in the mineral core, seems highly unlikely unless large (> 100 µm), defect-free gem quality garnets are involved. In addition, the calculations are based on anhydrous conditions. The manuscript points out that under hydrous conditions garnet will undergo markedly increased dissolution creep greatly promoting diffusion, increasing further that the likelihood that a protogenetic mineral would record the age of diamond formation.

Because of the opportunity to expand the body text we decided to move the discussion with respect to syngenetic vs protogenetic growth from the supplementary info into the main text. We have modified the text where we discuss instantaneous diffusion within the fluids and the fluid/melt interaction with silicate phases along the lines explained above.

3. I found the symbols in figure 2 a little confusing. The top panel has 10 blue coded inclusions and 9 green coded inclusions, but panel 2 and 3 have only 8 blue and 8 green. There are 26 inclusions in each panel so some inclusions of each of the age peaks are converted to grey in panels 2 and 3. The justification for this is not clear to me.

In order to group the garnet inclusions into genetically related populations that would potentially form an isochron, we use both the calculated depleted mantle model ages and the CaO content of the garnets (i.e. a threshold of 2.5 wt.% as indicated in Fig 2b and explained in the text). We find that by using a value of 2.5 wt.% CaO as a discriminant we obtain a grouping that is also coherent in terms of the trace element characteristics. This approach is also supplemented with the carbon isotope composition of the host diamond and REE of the inclusions. Two samples within the 1.1 Ga model age group become grey in Fig 2b because of their CaO content being lower than 2.5 wt.% (V306 and V471). They also have very different trace element contents and patterns (extremely steep HREE; see Table S.1 Fig S3) suggesting potential formation from a distinct type of fluid. In Fig. 5 these two sample plot within the fluid metasomatism field, rather than in the melt metasomatism field as is the case for the majority of the 1.1 Ga group samples. Similarly the sample in the 3.0 Ga model age group that has > 2.5 wt.% CaO and is thus excluded from the group in Fig 2B (V526) has a significantly higher HREE content compared to the other samples from that group. This rigorous approach to group samples is designed to avoid the potential introduction of error into the isochron ages by including genetically unrelated samples. Hence, even though the excluded samples plot on or near the two isochrons in Fig 4, we believe that a scientific justification is required to include samples in a isochron population. The manuscript implicitly criticises previous studies that pooled many inclusions and hence we believe a rigorous methodology is required. To clarify the strategy used in grouping the inclusions we have expanded the section and improved the caption of Fig 2.

4. I would like to see some discussion of the samples that do not fall within the 2 main age peaks. Are these bad data? Or was diamond formation going on all the way from 4 Ga to 1 Ga?

The current data-set indeed suggests diamond growth during multiple stages potentially from 4.1 to 1 Ga. More data are required to establish the significance of garnets that record older and younger ages than the 3 Ga event. As requested by both reviewers (see next section) and because of the less strict word limit for Nature Communications, we have added a paragraph to discuss the samples that are not included in the isochrons (end of the garnet geochemistry section).

5. Although the single inclusion data is very impressive it is a shame that there is not more information on the growth zones from which the inclusions were extracted. The latter approach was done, very effectively, by Suzette Timmerman from the same group. Simply using the surface features of the diamond to classify as inside or rim (Table S2) is rather crude. By losing the context of the inclusions it is not clear whether each individual diamond is from a single growth event (as implicitly assumed here) or whether there are diamonds with cores and rims of different ages. Diamond V683 has very different $\delta^{13}C$ for inside and rim implying that this sample may have a core and overgrowth of very different ages. FTIR on the fragments would have been helpful (see point 6).

We agree with Simon that a spatially resolved *in situ* approach, where the location of the inclusion with respect to the diamonds growth zones would be monitored with respect to the carbon and potentially nitrogen isotopes, would have been ideal as this would allow resolving multiple growth events within single diamonds. Unfortunately, this was impossible in this study due to the fact that the diamond inclusions were already extracted from the diamonds before they arrived at the VU. The extremely rare, relatively large peridotitic inclusions (> 100 μ m), had been liberated by Jeff Harris and co-workers at Muenster in the 90s. FTIR analyses were undertaken on unpolished diamond fragments during a visit of M.U.G. to the University of Alberta. Unfortunately data are of very poor quality owing to diffraction and reflection of IR beam on the fractured and uneven sample surfaces. Furthermore, the nitrogen contents are generally low, so errors are proportionally large. Therefore although we totally agree with the reviewer, we are unable to obtain high quality FTIR data. All

we were able to determine from the FTIR data was that the diamonds have relatively high N aggregation and hence are not young. Unfortunately this does not add significantly to the manuscript so we are unable to expand the discussion in this context.

6. I find it fascinating that the 2 periods of growth identified here by radiometric dating are remarkably similar to the dates of growth deduced by FTIR for Murowa, if it is assumed that the temperature experienced by Murowa diamonds was constant throughout their history in the mantle (note the important caveat!). In our recent paper on FTIR techniques we used a Murowa diamond as an example of 2 stage thermal modelling (Kohn et al., Lithos 265 148-158). The scenario explored in our figure 8 has initial diamond growth at 3.2 Ga, an overgrowth at 1.1 Ga and exhumation at 500 Ma. This is remarkably similar to the scenario proposed here for Venetia and considering that the two localities are only 200-300 km apart I wonder if they are both affected by the same regional events.

With our current data we cannot distinguish if the two observed episodes of diamond growth are recorded in single crystals by core and rim growth zones. The carbon data on core and rim chips suggests homogeneity for single diamonds except for sample V683 as pointed out by Simon above. A new study on diamond plates from Venetia in which we hope to study gems with garnet inclusions in multiple growth zones is planned to resolve if multiple growth is indeed recorded in single diamonds as was inferred based on the FTIR work on Murowa diamonds by Kohn et al., 2016.

*Simon Kohn
University of Bristol*

Reviewer #2

*Review of NCOMMS-17-03711-T
Nature Communications*

Title: Archaean and Proterozoic diamond growth from contrasting styles of large-scale magmatism

Authors: Janne M. Koornneef, Michael U. Gress, Ingrid L. Chinn, Hielke A. Jelsma, Jeff W. Harris, and Gareth R. Davies

The manuscript reports dates derived from individual garnet inclusions from 26 diamonds, sourced from Venetia, South Africa. Sm-Nd model ages derived from individual garnets represents a relatively new technological capability for isotopic dating. In recognizing two model age clusters, the authors divide the garnets into an older 3 Ga group and a younger 1.1 Ga group, which are further distinguished by their Ca content and the carbon isotopic composition of the host diamond. The two groups also have slightly different rare-earth element patterns. One of the more compelling observations is that the two groups of garnets each appear to conform to well-behaved Sm-Nd isochrons.

The authors stress that a previously reported isochron age obtained from measuring composite/pooled garnet suites (Richardson et al. 2009) is incorrect. From here the connection is drawn between the two newly-defined age groups (2.95 Ga and 1.15 Ga) and contemporaneous large-scale crustal magmatic events. Trace element characteristics of the two groups are used to argue that the 2.95 Ga garnets record diamond growth from "cool" C-O-H fluids, while the 1.15 Ga garnets record diamond growth related to melt metasomatism.

Overall, the resulting data are of good quality and the implications are interesting. However, the presentation of the derived ages (especially the isochrons) could be strengthened and the large-scale implications require further development. The case to back up the isochrons could be made more thoroughly, which would make for a stronger paper.

We appreciate the reviewer's positive comment about the quality of our data and the interesting implications.

Following the reviewer's recommendation to make a stronger case for the link between the isochron ages and the large scale interpretation we have expanded the text in the respective sections (see track changes document and more detailed comments below).

It would also be helpful to make some brief mention or interpretation of the 7 garnets whose model ages don't fit nicely into the two modes.

We have added a short paragraph to discuss the 7 garnets for which model ages do not fall within the specific age groups. This point was discussed above with respect to the similar request made by Simon Kohn.

Currently too much discussion is devoted to pointing out the "incorrect" age obtained from pooled garnet analyses (Richardson et al. 2009). These previous measurements were made quite meticulously and are not really "incorrect," so much as they are misinterpreted. It is obvious that pooling samples can give an average measurement, but these previous measurements are still in total agreement with the new data and can now be re-interpreted more accurately. More could be gained by embracing the old data rather than refuting it.

We accept that the previous measurements made on the pooled inclusions suites have indeed resulted in good quality data; however the age constraints based on such pooled inclusion suites are now confirmed to be incorrect and this is not a point that can be avoided in the manuscript. Having said this, we agree to use more careful language and have used the specific suggestions made by the reviewer to change the wording from 'incorrect' to 'misinterpreted' and 'ages with little meaning'. See the specific comments below and the track changes in the document.

The main area with room for improvement lies in the connection between diamond formation and tectonic/magmatic activity in the crustal record. The title and abstract make such claims boldly, but the text falls short of making a concrete connection. The parallels between garnet inclusion ages and large-scale magmatism are oversimplified. Nevertheless, the data and the potential for understanding large-scale processes recorded at the top (crust) and bottom (diamonds) of the lithosphere are very intriguing. If a case is to be made for strong connection between diamond formation and large-scale magmatism, a more detailed discussion, with geological map and perhaps cartoon model, should rightly be included.

Because of length limits for the initial submission to Nature Geoscience the discussion of the link between tectonics and diamond formation was kept brief and simple. We have expanded and substantiated the discussion with respect to the implications for the relations with large scale magmatic activity. Furthermore we have now included an additional figure including a map and a cartoon to place the implications into a geological and tectonic framework; i.e. extension of the Zimbabwe Craton margin at ~3.0 Ga and the large igneous province at 1.1 Ga. Our line of reasoning follows similar arguments made by previous workers who related diamond forming events to regional geological events; see reviews in Shirey et al (ref 1); Richardson et al. 2009; Venetia ref 13). This point was also made in the introduction of the original manuscript; Inclusion-bearing diamonds derived from the sub-continental lithospheric mantle (SCLM), preserve evidence of tectono-thermal events such as continental assembly and mantle melting^{1,3-5}. This section is retained in the current manuscript. More specific comments:

-Since line numbers were not included (apologies for that bad oversight), the comments below will refer to page number, and line number counting down that particular page.

Most of the following more specific comments were readily addressed within the track changes document. We only reply to the comments where we think we need to clarify the strategy for the changes we made or in a very few cases chose not to make.

Title (p1): A title revision is suggested, following revisions to the manuscript. It may be more helpful to note the actual numerical ages determined, and specifying garnet inclusions and the locality.

Because of the 15 word limit set for the title we are afraid that the request for the title to include both the ages, the locality and garnet inclusions will be impossible to meet. If the editor would allow the title to be 2 words over the limit, we propose to change the title to:

“Diamond growth at 3.0 and 1.1 Ga beneath Venetia, South Africa from contrasting styles of large-scale magmatism” . Otherwise, we would be happy to slightly modify the original title, with the addition of the locality name as also suggested by Simon Kohn. We await to hear the editors decision on this.

P2, line 4: state the number of diamond samples; done

P2, line 9: consider changing two billion to 1.8 billion to avoid giving the reader the appearance of exaggeration; done

P2, line 12: consider changing “incorrect” to a different word, such as a “misinterpretation”, since the pervious data are still correct. The incorrect part was the underpinning assumption that the collection of Venetia peridotitic diamonds were of a single age. done

P2, line 13: missing “The” before “results” done

P3, line 4: Here it is stated that diamonds form from supercritical C-H-O-S fluid, which is consistent with popular ideas in the current literature, but later in the text diamond growth from melt is discussed, which seems inconsistent. Granted, this is not fully resolved in the literature, but as it is written, this inconsistency might be confusing to readers.

To avoid the apparent confusion we have added the term ‘melt’ to the indicated section. “Diamonds crystallise from metasomatic reactions involving C-H-O-S-rich supercritical-fluids and/or melts and can include minerals that..”

We have not expanded the text in a major way and tried to avoid unnecessary discussion of this subtle point. A supercritical fluid contains a major silicate fraction so using the term melt or fluid is somewhat semantic. If required by the editor we will add such a sentence but currently chose not to do so to maintain the flow of the manuscript.

P3, line 4: remove comma; done

P3, line 6: it is important not to label kimberlites as the sole bearer of diamonds, since related volcanic rocks such as lamproite also transport diamonds; done

P3, line 5-7: Here and in a few other places, sentences need some grammatical nit-picking (Here the verbs are not in parallel form: “are transported” versus “protect”)

We agree and have changed lines 5-7 to “Diamonds, transported from the SCLM to the surface as xenocrysts in kimberlitic or related magma types, protect the mineral inclusions from secondary processes such as later mantle metasomatism and re-equilibration with the host magma.”

P3, line 9: “fundamental information about the evolution of the Earth’s interior” sounds a little vague

To be more specific, we edited the text, which now reads “fundamental information about the tectono-magmatic processes that led to the formation and modification of the lithospheric keels that underlie the oldest parts of the Earth’s continents”.

P3, line 10: add “lithospheric” before “diamonds”

We agree that the specific distinction between ultra-deep and sub continental lithosphere diamonds is required at this point and have edited the text appropriately.

We added: derived from the sub-continental lithosphere

P3, line 14: be more specific than “low elemental abundances” (trace element concentrations?); done

P3, line 19-20: “a previous study... studied” ; done

P3, line 21: change “combined” to “divided”; done

P4, line 3: change “conclusion” to “interpretation”; **done**

P4, line 10: the assertion that the inclusions “directly date” magmatism might be a bit too bold. Consider just “date”; **done**

P4, line 13: consider change “processes controlling diamond formation” to “processes associated with diamond formation”; **done**

P4, line 16: consider adding an average inclusion diameter, since the weight is not easy for most readers to interpret or visualize (e.g. 0.2-0.4 mm garnets); **done**

P4, line 21: This statement only mentions fluid. What about the diffusion through the solid minerals? Reader may wonder if diffusion from fluid-mineral interface into the interior of a solid mineral is also rapid. Is the garnet above “closer temperature”?

This point is similar to that made by Simon Kohn. Hence we refer here to our reply above. As indicated there, we have moved the discussion from the supplementary materials into the main text and have included discussion of the rate of diffusion in garnet and the timescales this implies. The point raised by the reviewer was correct and by using the additional space available, this issue is now addressed in detail.

P5, line 1: specify if the flat crystal surface was level (i.e. ideally surface is both flat and perpendicular to beam); **done**

P5, line 1: delete comma; **done**

P5, line 6: new sentence doesn't flow from previous sentence

As suggested a link between the sentences has been added to improve the flow of the text.

P5, line 13-14: carbon isotope values are missing minus signs; **done**

P5, line 16: consider changing “sigmoidal” to “sinusoidal” **done**

P5, line 17-18: description of REE pattern shape could be more clearly written.

The text has been edited and we believe the description is now clearer.

P5, line 19: the isochrons are impressive and could be discussed in more detail and mentioned in abstract

We thank the reviewer for the suggestion to emphasize the isochrons more; we have expanded the text and now mention the isochrons in the abstract. We have also added more emphasis to the data but find it hard to praise the quality our data even more and prefer an understated approach.

P5, line 20: Consider changing “no relationship” to “no clear relationship that would indicate simple two-component mixing”. There is still some discussion here that should be done to increase the reader's confidence in the isochrons.

We have expanded the discussion to point out the good agreement with the model ages and the isochrons. We also stress the lack of the two component mixing relations, which further supports the significance of the isochron ages. We have also added a comment as to why we expect the model ages to be correct in harzburgitic garnets in contrast to the multi-phase mineral assemblages that occur in lherzolithic and eclogitic inclusion populations.

P6, line 2-16: This section can be shortened significantly. Also, the mention of Premier and Udachnaya pooled ages should appear here in this section rather than in the final sentence of the paper.

In accordance with the reviewer's suggestions we have moved the section that refers to Premier and Udachnaya into this paragraph. We do, however, not agree that this section can be shortened significantly, especially as this contrasts with the reviewer's request to make more of the data. In light of the opportunity to expand the manuscript for publication in Nature Communications we decided to keep the approximate length by

emphasising the potential implications of our data for the re-interpretation of previous published studies and the regional controls of diamond formation.

P6, line 4: consider changing “incorrect” to “average” or “ages with little real meaning” done

P6, line 8: change “published data” to “composite inclusion data” or “pooled inclusion data” done

P6, line 12: the association between 1.1 Ga diamond growth and “basaltic” magmatism appears abruptly here, for the first time, without rationale.

This discussion has been expanded

P6, line 15: consider changing “unconstrained” to “unrecognized” done

P6, line 19: “The different geochemistry of garnet inclusions” should be worded more clearly to specify the two age group

We edited the text as suggested; We added “major and trace element” to be more specific. Note that this sentence is an introductory sentence to the rest of this section where the distinct geochemistry of the two age groups is described in more detail.

P6, line 22: it would be helpful to explain briefly why Fig 5 indicates melt or fluid dominated metasomatism. This is important in developing the rest of the story and making connections to magmatism.

We have added a sentence to explain the effect of melt and fluid metasomatism on the trace element signatures; i.e. ratios of highly to mildly incompatible elements decrease strongly in garnets resulting from fluid- to melt metasomatism. The reference to the original work is also included.

P6, line 23: “low HREE” –should this say “high HREE”?

This comment is incorrect. The older group samples have relatively low HREE; hence no specific changes made.

P7, line 3: why does LREE enrichment indicate so specifically a “C-H-O medium highly enriched in trace elements”

The observed LREE enrichment observed in the garnet inclusions is despite the major element depleted harzburgitic characteristics. This requires that the medium from which the diamond grew (a C rich fluid) must have been enriched in LREE. We adjusted the text to clarify this point.

P7, line 5: “elevated ... HREE” –should this say “lower HREE”?

This comment is incorrect. The 1.1 group samples have relatively high HREE; hence no specific changes made.

P7, line 7: “melts derived from the asthenosphere” sounds vague

We have expanded the discussion text and now go into more detail about the nature of the asthenospheric melts in the section about the 1.1 samples. This sentence, which is inferred to sound vague, is the introduction to the more detailed discussion about the inferred tectonic events. We have added a bridging sentence to the more detailed section that follows to make it clear that we are starting a more detailed discussion of the regional implications of the data.

P7, line 10: “fluids mobilized by crust-forming magmatism” sounds vague

P8, line 2: More explanation is needed here in the inferred connection between contemporaneous tonalite plutons and diamond growth.

We have expanded the text with respect to the relation between the crust forming magmatism and diamond growth to meet the request of the reviewer. As explained above and below, our line of reasoning follows similar arguments made by previous workers who related diamond forming events to regional geological events; for example, see reviews in Shirey et al (ref 1); Richardson et al. 2009; Venetia ref 13).

P8, line 3-6: consider rewording this sentence to make it easier to read. Text has been edited

P8, line 7: sentence truncated at “and” edited

*P8, line 9-10: Is this fractionation mechanism valid in light of recent work suggesting that mantle harzburgite is unable to buffer significant O₂ required support redox-reaction diamond formation models in the SCLM? (Luth, R.W., Stachel, T., 2014. The buffering capacity of lithospheric mantle: implications for diamond formation. *Contrib. Mineral. Petrol.* 168, 1083.)*

Although the question posed by the reviewer and work by Luth and Stachel is interesting, we consider that an extensive discussion of the topic to be outside of the scope of this manuscript. Luth and Stachel developed a model to evaluate the buffering capacity of the mantle during diamond precipitation, however, as they admit in their paper, their approach is simplified (i.e., only ferrous-ferric equilibria are considered) and the precipitation process is a function of multiple unknown variables (composition of the fluid, and P-T conditions and changes therein) that make an accurate quantitative evaluation tricky. Note, however, that their statements with respect to a relation between CaO poor harzburgitic inclusions and the presence of diamonds owing to the high solidus temperature of depleted residues is relevant to our work. Our conclusion that we require a reduced fluid that becomes oxidized to precipitate the 3.0 Ga diamond with CaO poor, depleted inclusions fits both with the sinusoidal trace element patterns and with a mechanism to fractionate the carbon isotopes. In this light we infer that the diamond precipitation could reflect percolation of C saturated reduced fluids along a geotherm. We have added a sentence to point out this consistency/relationship.

P8, line 12: How is diamond formation directly linked to the Umkondo LIP? Is their age overlap and broad mantle-derived geochemical similarity enough to say they are directly linked? More justification and explanation would be helpful here.

We consider the age overlap and geochemical similarity between the Umkondo LIP products and garnet inclusions of the 1.1 group striking and thus are indeed of the opinion that we can infer a relationship. Currently we have no further justification to link the diamond growth and large-scale magmatism. We have taken out the word ‘directly’ to tone down the inference. We have to admit that we have followed common practice in making comparison between tectono-magmatic events/ Os depletion ages from mantle xenoliths and diamond formation (see review by Shirey et al. ref 1 and refs 3-6; 9-12). A similar approach, for example, was also taken in the original study of the Venetian diamond inclusions by Richardson and co-workers (13). The basic argument is that a major event is required to move the large amounts (>> M tons) of carbon needed to produce the diamond suites. For example, the Venetia kimberlite cluster only comprises a surface area of 28 ha. Since opening in 1992, the mine has produced ~10 M carats per year and the total resource is estimated to have contained several 100 M carats, the majority of which are harzburgitic. We have added a short discussion of this point to the paper to stress the need for large scale processes to produce many tons of diamond.

P8, line 17: Explain briefly why the garnets “imply an origin related to high-T melt metasomatism”

In addition to the link to the Stachel et al. paper, we discuss the observation of the relation between HREE content and T; i.e. garnets with high HREE record high T.

P8, line 23: Explain why a “major thermal perturbation” is required

We edited the wording from “required a major thermal perturbation” to “was associated with a major thermal perturbation”. Elsewhere in the manuscript the link between higher diamond formation temperatures and composition has been developed. We believe that further discussion here would now lead to unnecessary repetition.

P9, line 2: Why is the magmatism claimed to be “rift-related” here? Rifting was not mentioned earlier in the discussion of tonalite plutons

The rift-relation of the crust forming magmatism was put back into the discussion above to re-establish the link that was accidentally deleted during a previous re-edit to shorten the text to fit with the Nature Geoscience word limit.

P9, line 2-4: This sentence could use clarification

We have rephrased the sentence to clarify. We have also added another line at the end of the manuscript to emphasise that further investigations on inclusions and their host diamonds are required to establish if the observed change in the style of diamond growth is recorded over a larger geographical area and potentially world-wide.

P9, line 6: change “obscures” “obscure” done

Figure 1: This figure is not absolutely necessary and could be moved to supplement, making room for some kind of geological map/model.

Because Nature Communications allows 10 display items we have decided to keep the figure and added an extra figure to support the relation between diamond growth and large-scale magmatism.

Figure 2: Should the legend/symbol for “all data” be called “remaining” or “unassigned” or “outlier” data? The collection of grey diamonds is not all the data. Same goes for Fig 5.

We have changed the symbol legends of Fig 2 and Fig 5 to label the “Remaining data” as suggested by the reviewer.

Figure 2 caption: Consider changing “more depleted” to “lower.” Are the carbon isotope values shown average measurements for each diamond?

We have plotted carbon isotope data for fragments from the interiors of the crystal as we consider it more likely that those are representative for the inclusion host compared to the rims. Note, however, that the carbon isotope data were homogeneous for cores and rims for all but 1 sample. We now specify in the caption that we are reporting the interior of the diamonds.

Figure 3 caption: The caption says blue is the 3.0 Ga group, but the legend says blue is the high Ca group (and green is 1.1 Ga, low Ca). This does not agree with Figure 2.

The error in the caption was corrected.

Figure 4 caption: Refers to symbols in Fig 1, but shouldn't it say Fig 2?

Yes, we apologise for this error and have made the required edits.

References, ref 19: incomplete; edited

Supplementary Information:

-How were inclusions chosen for the study?

-How were the inclusions removed from the diamonds?

-Some physical description of the diamonds themselves should be given. If there are really two major age groups, it would be interesting to note if there are any differences in the diamonds, aside from the carbon isotopes (e.g. morphology, nitrogen concentration, nitrogen aggregation, internal growth zonation).

To address all three questions above we have added a section to the start of the methods to describe where the samples came from and that they were already liberated from the hosts at the onset of this study, which unfortunately inhibits relations between the age groups and diamond morphology or geochemistry of individual growth zones to be examined.

P2, line 9: do not abbreviate concentrated to “conc.” done

P4, line 5: change “inclusions formed” to “inclusions are formed” done

P5, line 7: should this refer to Fig S3 rather than Fig 2?

The structure of the edits makes this comment (although correct) redundant.

P6, line 2: "Here we date individual garnet inclusions..." this sounds out of place, like it belongs in the introduction to the main text

We agree and have removed this section.

Figure S2 caption: For symbols shouldn't it say refer to Figure 2? For the "small grey diamonds," consider changing to a different symbol, since grey diamonds are already used, and mention here that these additional data points are for Venetia garnets.

We edited the caption to figure 2. We have, however, not changed the small grey diamond symbols. We consulted colleagues and they agree that the grey diamonds are clearly distinguishable from the other symbols in the figure.

Figure S3: this figure is not referred to in the text

The structure of the edits makes this comment (although correct) redundant.

*Table S1: state units (wt %, ppm) **done***

*Table S3: provide reference for Venetia age of 520 Ma; **added***

REVIEWERS' COMMENTS:

Reviewer #1 (Remarks to the Author):

I am satisfied that the authors have taken all the comments of the reviewers seriously and that the revised manuscript is now suitable for publication without further changes.

Reviewer #2 (Remarks to the Author):

Manuscript#: NCOMMS-17-03711A

Title: Archaean and Proterozoic diamond growth from contrasting styles of large-scale magmatism beneath Venetia, South Africa

Dr. Koornneef and co-authors have made very good revisions to the manuscript. Concerns and suggestions of both reviewers appear to have been carefully considered and properly addressed according to the detailed revision notes. In my opinion, the manuscript should now move forward toward publication.

Point by point response to reviewers

REVIEWERS' COMMENTS:

Reviewer #1 (Remarks to the Author):

I am satisfied that the authors have taken all the comments of the reviewers seriously and that the revised manuscript is now suitable for publication without further changes.

Reviewer #2 (Remarks to the Author):

Manuscript#: NCOMMS-17-03711A

Title: Archaean and Proterozoic diamond growth from contrasting styles of large-scale magmatism beneath Venetia, South Africa

Dr. Koornneef and co-authors have made very good revisions to the manuscript. Concerns and suggestions of both reviewers appear to have been carefully considered and properly addressed according to the detailed revision notes. In my opinion, the manuscript should now move forward toward publication.